# Sorption of Phosphate on Douglas Fir Biochar Treated with Magnesium Chloride and Potassium Hydroxide for Soil Amendments

**Beatrice Arwenyo** [1,2] **, Chanaka Navarathna** [1] **, Naba Krishna Das** [1] **, Addie Hitt** [1] **and Todd Mlsna** [1,*]

[1]   Department of Chemistry, Mississippi State University, Mississippi State, MS 39762, USA
[2]   Department of Chemistry, Gulu University, Gulu P.O. Box 166, Uganda
*    Correspondence: tmlsna@chemistry.msstate.edu; Tel.: +662-325-6744; Fax: +662-325-1618

**Abstract:** With increasing climate variability, a sustainable crop production approach remains an indispensable concern across the globe. In this study, P retention/availability of $MgCl_2.6H_2O$/KOH modified Douglas fir biochar was assessed. The $MgCl_2 \cdot 6H_2O$/KOH treated Douglas fir biochar was prepared by sequentially treating Douglas fir biochar with magnesium chloride and potassium hydroxide solutions. The biochar's surface area, pore volume, morphology, and elemental compositions were determined using BET, SEM, SEM/EDS, and powder X-ray analyzes. Both surface area and pore volume were reduced by more than 97% following modification. Similarly, the morphology and elemental compositions changed after modification. The maximum P adsorbed corresponding to Langmuir–Freundlich model was 41.18 mg g$^{-1}$. P sorption on biochar soil mixture was pH dependent. More studies are required to establish the field applicability of P-laden $MgCl_2 \cdot 6H_2O$/KOH-modified Douglas fir biochar as a soil additive.

**Keywords:** modified biochar; soil pH; P retention and P availability





## 1. Introduction

Globally, fertilizer and manures are the primary sources of phosphorus (P) for improved crop yield. However, the constant use of fertilizers and manure for food and fiber production is considered the primary cause of eutrophication and its related problems [1,2]. Eutrophication is associated with algae blooms and the insufficiency of oxygen in the water for aquatic animals. Furthermore, in such anaerobic conditions, methanogens decompose organic matter into methane, a gas with higher contributions to global warming than carbon dioxide per unit [3]. The United State Environmental Protection Agency (USEPA), in an attempt to minimize eutrophication, has set limits for total phosphate concentrations at 0.05 mg L$^{-1}$ and 0.1 mg L$^{-1}$ in streams entering lakes and flowing water bodies, respectively [4,5].

Biochar, a solid by-product formed from the anaerobic combustion of various biomass, has caught the attention of many researchers due to its multiple environmental benefits [6]. Recently, biochar's ability to adsorb environmental pollutants has been reported [7]. Moreover, it is known that the sorption capacity of biochar can be improved by modification [8]. Common treatment agents include; oxidants (HCl, $HNO_3$, $H_2O_2$, $H_3PO_4$), reducing agents (NaOH, KOH, and $NH_4OH$), and metal salts ($Fe^{3+}$, $Mg^{2+}$, $Al^{3+}$, $Zn^{2+}$, $Mn^{2+}$ and $Ag^+$). However, for soil amendment, phytotoxicity due to excess $Fe^{2+}$, $Al^{3+}$, and $Mn^{2+}$ as treatment agents may occur [9,10]. Excess $Fe^{2+}$ in plant cells accelerates redox reactions by acting as an electron donor [11]. $Al^{3+}$ inhibits root elongation, root hair growth, lateral root development, and rhizobial infection of the roots [12], while $Mn^{2+}$ causes phytotoxicity by producing reactive oxygen species and interfering with the metabolism of essential metals [13]. In addition, P sorption by clay minerals, $Ca^{2+}$, and oxides of iron or aluminum in soil-forming insoluble complexes and precipitates has

been extensively documented [14,15]. Akgül et al. [16] observed that biochar modified with $Mg^{2+}$ had higher sorption capacity for $PO_4^{3-}$ in comparison to $Fe^{3+}$, $Al^{3+}$, and $Mn^{2+}$ ions. Additionally, other studies indicated that the adsorbed phosphate could be released in soil for plant uptake as nutrients [17,18].

Modified P-laden Douglas fir biochar, made from the sequential treatment of biochar with magnesium chloride, potassium hydroxide, and aqueous potassium phosphate solution, can be a cheap and eco-friendly alternative for soil P management. Moreover, in addition to P, the P-laden modified Douglas fir biochar could supply the soil with other essential plant nutrients, including Mg and K. To date, however, information on P retention/availability of modified Douglas fir biochar as a soil additive is still scanty. In addition, despite several studies on the sorption of Phosphate by modified biochar, commercial biochar such as Douglas fir biochar has received less attention regardless of its low cost and availability. Therefore, this study evaluated the P retention/availability of modified Douglas fir biochar for its potential use as a soil additive. The study estimated the pH range over which modified Douglas fir biochar sorbs phosphate, determined the P sorption capacity of magnesium chloride/potassium hydroxide modified Douglas fir biochar, and examined the sorption of P by modified Douglas fir biochar/soil mixture.

## 2. Materials and Methods

### 2.1. Source of Biochar Used

Biochar used for this study was derived from Douglas fir as a by-product of waste wood gasification to syngas. Raw Douglas fir chips (~3inch lengths) were auger-fed into an updraft gasifier for a ~1 s residence time at about 900–1000 °C. The resulting biochar was carefully washed several times with water to remove ash and other impurities before drying in air. The dried biochar (DFB) was ground, sieved through 50 mm mesh, and stored in closed vessels for further use.

### 2.2. Preparation of Modified Biochar (MBK)

Modified biochar was made by sequential mixing and drying of Douglas fir biochar (DFB) with solutions of magnesium chloride and potassium hydroxide, respectively. The choice of both magnesium chloride and potassium hydroxide as treatment agents for biochar modification was because $Mg^{2+}$ is known to have good P sorption capacity [19]. In addition, both $Mg^{2+}$ and $K^+$ are essential to plant nutrients [20]. Although Fe, Mn, and Zn are useful for plant growth, they are micronutrients that are needed by plants in small quantities [21]. A substantial quantity of $Al^{3+}$ and $Mn^{2+}$ in the soil is toxic to plants. Al–toxicity inhibits root growth by altering root membrane structure and functions [22], while Mn-toxicity restricts shoot growth through metabolic alterations [23]. Moreover, phosphate deficiency in acid soils is linked to elevated amounts of $Al^{3+}$, $Mn^{2+}$, and $Fe^{3+}$ [24].

Depending on pH, phosphate sorption on MgO-modified biochar's surface "s" may involve complexation [25], precipitations [17], and electrostatic interactions [18].

$$MgCl_2(aq) + 2KOH(aq) \rightarrow Mg(OH)_2(s) + 2KCl(aq) \tag{1}$$

$$\text{Then} \quad Mg(OH)_2(s) + heat \longrightarrow MgO\ (s) + H_2O\ (g) \tag{2}$$

Complex formation
Mononuclear

$$sMgO - OH_2^+(aq) + H_2PO_4^-(aq) \rightarrow sMgO - H_2PO_4\ (s) + H_2O\ (l) \tag{3}$$

Binuclear

$$2sMgO - OH_2^+(aq) + HPO_4^{2-}(aq) \rightarrow s(MgO)_2HPO_4(s) + 2H_2O(l) \tag{4}$$

Trinuclear

$$3sMgO - OH_2^+ (aq) + PO_4^{3-} (aq) \rightarrow (sMgO)_3 PO_4 (s) + 3H_2O \ (l) \tag{5}$$

Precipitation

$$MgO \ (s) + H_2O \ (l) \rightarrow Mg(OH)_2 (aq) \rightarrow Mg^{2+} (aq) + 2OH^- (aq) \tag{6}$$

Then

$$Mg^{2+} (aq) + H_2PO_4^- (aq) \rightarrow Mg \ (H_2PO_4)_2 \ (s) \tag{7}$$

$$Mg^{2+} (aq) + HPO_4^{2-} (aq) \rightarrow MgHPO_4 (s) \tag{8}$$

$$Mg^{2+} (aq) + PO_4^{3-} (aq) \rightarrow Mg_3 (PO_4)_2 (s) \tag{9}$$

Electrostatic attraction
Also, protonation can occur.

$$MgO(s) + H_2O \ (l) \rightarrow \equiv Mg - OH^+ (aq) + OH^- (aq) \tag{10}$$

Then

$$MgOH^+ (aq) + H_2PO_4^- (aq) \rightarrow MgOH^+ - H_2PO_4^- (aq) \tag{11}$$

$$MgOH^+ (aq) + HPO_4^{2-} (aq) \rightarrow MgOH^+ - HPO_4^{2-} (aq) \tag{12}$$

### 2.2.1. Preparation of Magnesium Chloride Modified Douglas Fir Biochar (MB)

To a clean 2000 mL glass beaker, 200 g of previously ground and sieved (<2 mm mesh) Douglas fir biochar (DFB) was added. To the biochar, a solution of magnesium chloride (0.52 M) made by dissolving 84.7 g of $MgCl_2 \cdot 6H_2O$ in 800 mL of deionized water was slowly added with constant stirring until a uniform mixture was formed. The mixture was then left on a magnetic stirrer for 6 h at 24 °C. After 24 h of standing, excess solution was filtered, and the residue dried in an oven at 100 °C to constant weight and stored in airtight polythene bags for further treatment.

### 2.2.2. Preparation Modified Biochar (MBK) from MB

To MB prepared previously (in Section 2.2.1) in a 2000 mL glass beaker, 800 mL of 5 M solution of potassium hydroxide, prepared by weighing 280.5 g of potassium hydroxide pellets and dissolving in 1000 mL of deionized water in a volumetric flask was added with constant stirring until a uniform slurry was formed. The slurry was stirred for 6 h with a magnetic stirrer and left to stand for 24 h, filtered, and the residue dried in an oven at 100 °C to constant weight. The modified biochar formed (MBK) was stored in tight plastic containers at room temperature, about (24 °C) until use.

### 2.3. Characterization of Biochar

### 2.3.1. Surface Analysis

The surface analysis of Douglas fir biochar and modified biochar were determined by Brunner–Emmet–Teller (BET) nitrogen gas physisorption. Before the adsorption measurements, the samples were degassed for about 6 h at 180 °C. Briefly, 0.1 g of each biochar sample was used to obtain the surface area and pore size from $N_2$ isotherms at about 77.3 K with a MicroActive TriStar II Plus (GA, USA) Version 2.03. The specific surface area and pore volume were calculated using Dubinin–Astakhov equation [$\log(a) = \log(a_o) - D\log^n \left( \frac{P_o}{P} \right)$] and the density function theory [$W_o = \left( \frac{44000 \ a_o}{\rho} \right)$], respectively [26]. Where a = the quantity of gas adsorbed per unit mass of adsorbent (mol g$^{-1}$), $a_o$ = the micropore capacity (mol g$^{-1}$),

D = a constant, P and $P_o$ = the equilibrium and saturated vapor pressures of the adsorbate at temperature T(K), respectively,

$$W_o = \text{the limiting micropore volume (cm}^3\text{ g}^{-1}) \text{ and}$$

$$\rho = \text{the density of adsorbed gas (gcm}^{-3}).$$

Determination of Surface Morphology, Structural Chemical Composition, and Functional Groups

The morphology, surface textures, and qualitative elemental composition of DFB, MBK, and post sorption MBK here designated as P-enriched biochar (PEM) were examined with scanning electron microscopy (SEM) and SEM-EDX techniques at 5 kV using a JEOL JSM-6500F FE instrument (USA). Also, their structural and chemical compositions were analyzed by X-ray diffraction (XRD) using a SmartLab X-ray diffractometer (USA) by scanning 2θ from 0° to 90° at 1°/min. An XRD spectrum was obtained using the SmartLab X-ray diffraction system under the same conditions. The surface functional groups were studied by attenuated total reflectance Fourier Transformed Infrared Spectroscopy (ATR-FTIR) (ThermoScientific, USA).

### 2.3.2. Phosphate Adsorption Study

The phosphate stock solution (1000 mg P/L) was prepared by dissolving anhydrous potassium phosphate monobasic $KH_2PO_4$ (Sigma-Aldrich) in deionized water. The working solutions were obtained by diluting the stock solution with deionized water.

### 2.3.3. Determining the Point of Zero Charge ($pH_{pzc}$)

The $pH_{pzc}$ for MBK was determined using the solid addition method [27]. Briefly, 0.01 M NaCl aqueous solution with $pH_{initial}$ values varying from 4 to 12 was added to 0.05 g of MBK in a 50 mL polypropylene tube. The $pH_{initial}$ values were adjusted with either 0.1 M NaOH or 0.1 M HCl solutions. After agitating the tubes and their contents for 24 h at room temperature (23 °C), the supernatants were filtered using Whatman number 1 filter papers, and their pH was finally measured using a HI3221 pH meter. The $pH_{pzc}$ was obtained from the plot of $\Delta pH$ ($\Delta pH = pH_{initial} - pH_{final}$) against $pH_{initial}$.

### 2.3.4. Kinetic Studies

The batch kinetic study of phosphate sorption on MBK was performed by mixing 0.1 g of MBK in a 50 mL centrifuge tube with 30 mL of 100 mg P/L solution. The tubes were then shaken in a mechanical shaker at 200 RPM for varying time intervals. At a time, the tubes were withdrawn, and their contents filtered through Whatman filter paper number 1. The Phosphate concentration in the filtrate was determined by the ascorbic acid method using a UV spectrophotometer (model) at 830 nm wavelength.

The amount of Phosphate adsorbed at equilibrium (qe, mg P/g) was calculated using equation $q_e = \frac{(C_i - C_e)V}{M}$, where Ci and Ce (mg P/L) are the initial and equilibrium Phosphate concentrations, respectively, V (L) is the volume of the solution, and M (g) is the mass of the adsorbent (MBK).

### 2.3.5. Isotherm Studies

The adsorption isotherms were determined by mixing 0.1 g of MBK with 30 mL phosphate solution of concentrations 5, 52, 110, 220, 340, and 600 mg P $L^{-1}$ in a 50 mL centrifuge tube. The tubes were shaken for 48 h in a mechanical shaker (200 rpm) at room temperature (about 23 °C). Shaking was done for 48 h to ensure that reactions at all concentrations used reached equilibrium. After this, the samples were removed, filtered through Whatman filter paper number 1, and the corresponding phosphate concentration of the filtrate determined as in Section 2.3.4.

### 2.4. P Sorption by Biochar Soil Mixture

Adsorption by soil biochar mixture was examined to assess the efficiency of biochar amendments as a method for increased P retention in soil. To appropriate amounts of soil (initial pH = 5.3), 0.1, 0.15, 0.2, 0.25, and 0.3 g of MBK biochar were added in a 50 mL polypropylene tube to bring the total mass of soil/biochar mixture to 2 g and thoroughly mixed. To the mixture, 30 mL of 100 mg/L P was added, and the mixture was shaken at 200 rpm at room temperature in a mechanical shaker after adjusting pH to 4.0 and 6.5 with 0.1 M HCl or 0.1 NaOH. After 48 h of shaking, the content of the tubes was filtered using Whatman filter paper number 1, and the phosphate concentration of the filtrate determined as previously described in Section 2.3.4.

To examine P retention by soil/biochar mixture, 1.8 g of soil (pH = 5.3) was mixed with 0.2 g of biochar. 30 mL of 20, 30, 40, 50, and 70 mg/L P solution were added, and the mixture was shaken for 24 h in a shaker. The supernatant was then filtered, and the P concentration determined as above.

### 3. Results and Discussions

#### 3.1. BET Surface Analysis

The BET surface analysis results showed that both surface area and pore volume decreased after modification of DFB. The surface area was reduced by more than 99% (from 528.1025 $m^2$/g to 2.3942 $m^2$/g). Similarly, pore volume decreased by ~97% (from 0.041971 $cm^3$/g to 0.001141 $cm^3$/g). However, the adsorption average pore diameters increased from 2.7044 Å to 26.088 Å (Table 1). The decrease in the surface area following the modification of DFB is due to pore blockage by the modification agents and their aggregate. This incomplete blockage obstructs the passage of $N_2$ to micropores. According to Thi et al. [28], the reduction in the amount (volume) of nitrogen adsorbed by biochar manifests in the decrease in its surface area. This finding agrees with those of Fahmi et al. [29].

**Table 1.** BET Surface analysis of DFB, MBK, and PEM.

| Sample | BET Surface Area ($m^2 \ g^{-1}$) | Adsorption Average Pore Diameter (Å) | Total Pore Volume ($cm^3 \ g^{-1}$) |
|---|---|---|---|
| MBK | 2.39 | 26.08 | 0.001 |
| PEM | 137.23 | 12.72 | 0.048 |
| DFB | 528.10 | 2.70 | 0.042 |

DFB = untreated Douglas fir biochar, MBK = DFB treated with $MgCl_2 \cdot 6H_2O$ + KOH solutions, and PEM = MBK treated with $KH_2PO_4$ solution. The DFB pore diameter value may be uncertain due to $N_2$'s diameter being (~3 Å) higher than DFB's pore diameter.

#### 3.2. Surface Morphology, Structural Chemical Composition, and Functional Groups Determination

Scanning electron microscopy (SEM) and SEM/EDS (Energy Dispersive Spectroscopy) was performed to characterize DFB, MBK, and PEM. Corresponding micrographs and spectra are shown in Figures 1 and 2, respectively.

The SEM images of DFB, MBK (biochar after modification with magnesium chloride and potassium hydroxide), and PEM (MBK after sorption of phosphate) is shown in Figure 1. Before treatment, DFB had a distinguishable honeycomb structure with micropores. Consequently, DFB had the highest BET surface area compared to MBK and PEM. After modification and sorption of phosphate, however, the biochar surfaces became more heterogeneous, with crystal particles trapped on them. The decrease in the surface area demonstrated by MBK could be due to the infiltration of micropores in DFB by modification reagent resulting in the collapse of micropores and the subsequent creation of mesopores and macropores, which is manifested by the increase in pore diameter.

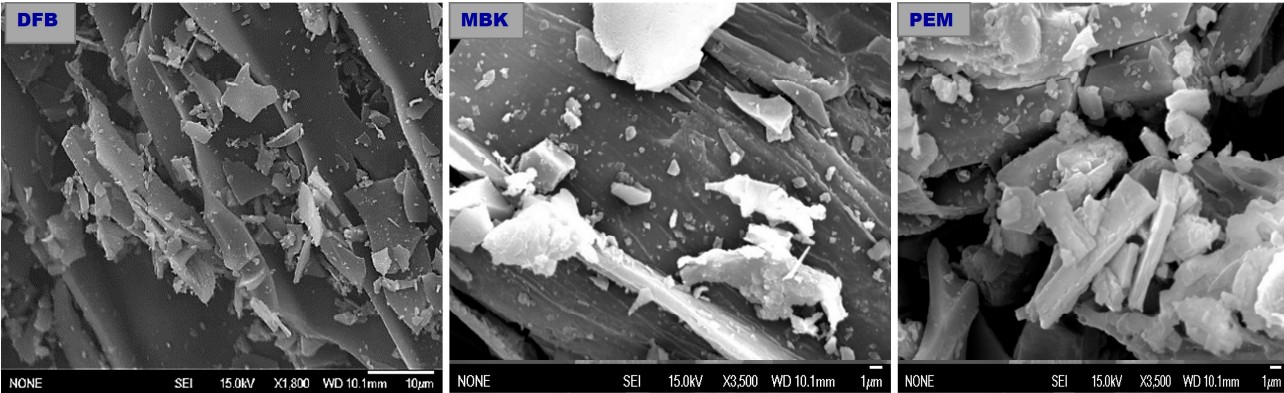

**Figure 1.** Scanning electron microscope (SEM) images of untreated Douglas fir biochar (DFB), DFB treated with $MgCl_2 \cdot 6H_2O$ + KOH solutions (MKB), and MBK treated with $KH_2PO_4$ solution (PEM).

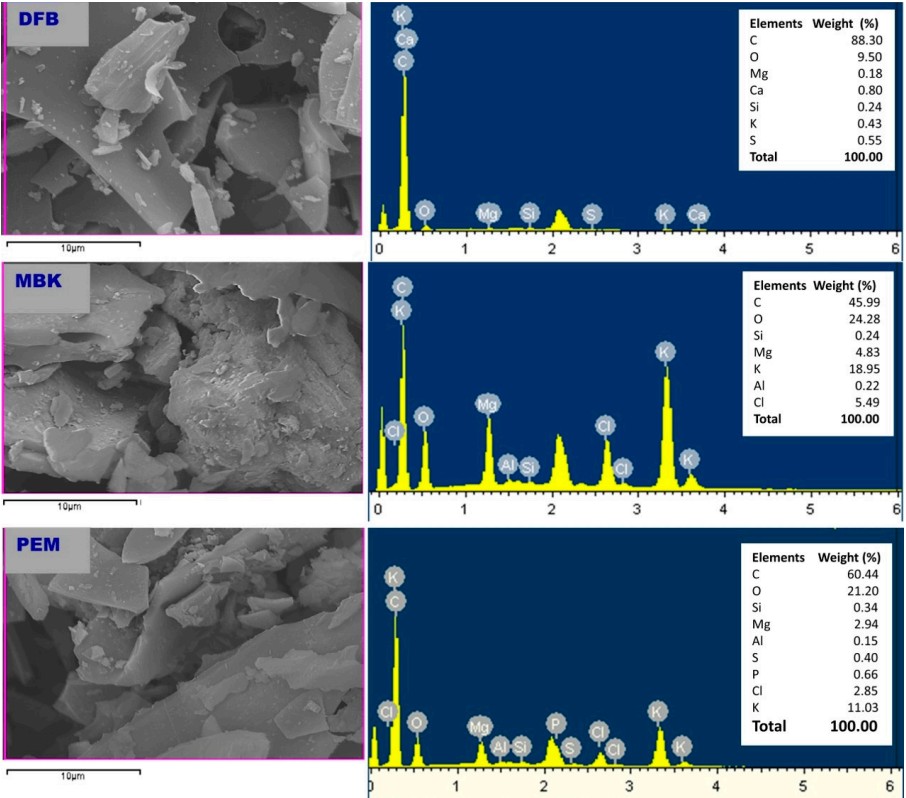

**Figure 2.** Energy-dispersive X-ray (EDX) spectra of untreated Douglas fir biochar (DFB), DFB treated with $MgCl_2 \cdot 6H_2O$ + KOH solutions (MKB), and MBK treated with $KH_2PO_4$ solution (PEM).

The SEM-EDX results (Figure 2) indicated that the main elements in DFB are C (88.3%), O (9.5%), Ca (0.8%), S (0.55%), K (0.43%), Si (0.24%), and Mg (0.18%). In addition to the major elements found in DFB, Cl (5.49%) and Al (0.22%) were also found in MBK. Furthermore, MBK had a higher amount of K and Mg than DFB, showing that Mg, K, Cl, and Al (impurity in modification reagents probably) were loaded to the DFB through the process of modification with magnesium chloride and potassium hydroxide. In addition to the main elements in MBK, PEM had P (0.66%), confirming its ability to adsorb phosphate.

In addition, the structural compositions of DFB, MBK, and PEM were examined with X-ray diffraction techniques (Figure 3). The peaks were identified by matching in qualX 2.0 Software. XRD patterns for DBF reveal the existence of two broad peaks like that of graphene between $2\theta$ = 20° to 30° and 40° to 50°. These peaks could be due to graphite diffraction [30,31]. Other sharp peaks showed mixed inorganic

components of calcite [32], quartz, lime, brucite, dolomite, and periclase [30]. New peaks emerged in MBK, providing evidence for the presence of MgO at $2\theta = 38°$ (1 1 1), $42°$ (200), $67°$ (3 1 1) [33] and $Mg(OH)_2$ at $2\theta = 7.96°$ (0 0 1), $13.79°$ (1 0 0), $15.94°$ (1 0 −1), $21.15°$ (1 0 −2), $24.01°$ (2 −1 0), $27.82°$ (1 0 −3), and $45.01°$ (3 0 2) [34,35]. Furthermore, additional peaks emerged in PEM due to hydrated $Mg_3(PO_4)_2$ at $2\theta = 4.65°$ (0 0 2), $6.16°$ (1 0 0) [36], and, $MgHPO_4$ at $2\theta = 4.65°$ (1 1 0), $5.73°$ (0 2 0), $6.80°$ (1 2 0), $13.18°$ (3 2 −1), and $16.51°$ (0 5 1) [33,36,37].

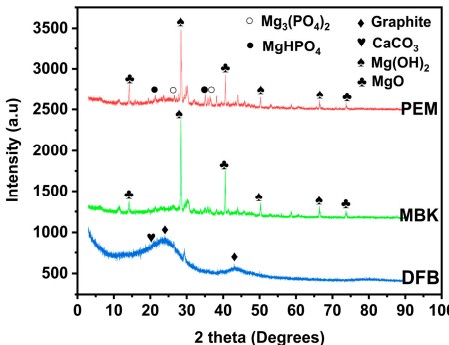

**Figure 3.** XRD spectra of untreated Douglas fir biochar (DFB), DFB treated with $MgCl_2 \cdot 6H_2O$ + KOH solutions (MKB), and MBK treated with $KH_2PO_4$ solution (PEM).

The ATR-FITR analysis of biochar sample surface functional groups is shown in (Figure 4). The spectra are comparable in wavenumber range, and the peaks occurred at similar wavelengths. The peaks showed that the biochar surfaces consist of mono-substituted aromatic and aliphatic hydrocarbons. The peaks were observed at wavelengths $3648.82$ cm$^{-1}$, $2925.44$ cm$^{-1}$, $2241.19$ cm$^{-1}$, $1448.84$ cm$^{-1}$, $1222.76$ cm$^{-1}$, between $600$–$900$ cm$^{-1}$ and $554.87$ cm$^{-1}$ corresponding to O-H (alcohol/phenol), V(C-H) vibration in $CH_3$ or $CH_2$, C=O stretch (aliphatic aldehydes), C=C stretch (aromatic), Aromatic C-O stretching, MgO, and P=O or P-O stretches, respectively [38,39].

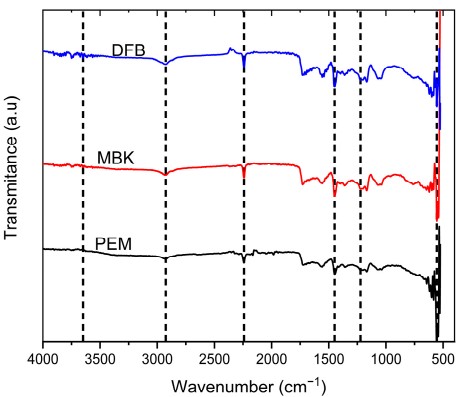

**Figure 4.** ATR FTIR Spectra of untreated Douglas fir biochar (DFB), DFB treated with $MgCl_2 \cdot 6H_2O$ + KOH solutions (MKB), and MBK treated with $KH_2PO_4$ solution (PEM).

### 3.3. Sorption of Phosphate on MBK

While adsorption describes a phenomenon in which solute particles attach themselves to the surface of an absorbent, adsorption kinetics refers to a curve displaying the rate at which a solute is retained or released from a solution to a solid phase interface for a given quantity of adsorbent at a particular temperature, flow rate or pH [40]. For this study, batch experiments were performed at room temperature (~25 °C) to determine sorption kinetic and isotherm.

### 3.3.1. Point of Zero Charge ($pH_{pzc}$)

The plot of $\Delta pH$ against the initial pH was used to determine the point of zero charges ($pH_{pzc}$) for MBK. The $pH_{pzc}$ defines the pH at which the overall charge on the surface of an adsorbent is zero [27]. For MBK, the $pH_{pzc}$ was ~10.3 (Figure 5), indicating that its surface would be positively charged for pH values below 10.3 but negatively charged for pH values above 10.3. According to Nguyen et al. [41], adsorption of cationic and anionic species is favored due to electrostatic attraction between the adsorbent and the sorbate when the adsorbent surface is negatively charged (pH > $pH_{pzc,}$) or positively charged (pH< $pH_{pzc}$). Therefore, for MBK, the adsorption of phosphate (anionic species) would be favored for pH values < $pH_{pzc}$ (10.3) because the positive charge surface of MBK would attract the negatively charged phosphate ions.

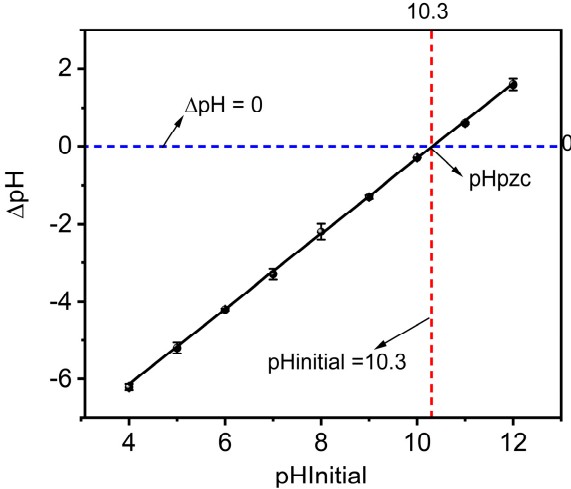

**Figure 5.** Point of zero charges of Douglas fir biochar treated with $MgCl_2 \cdot 6H_2O$ + KOH solutions (MKB).

### 3.3.2. Sorption Kinetics

Three models were used to describe phosphate sorption kinetics on MBK (Figure 6). The fitted pseudo-first order, pseudo-second order, and, Elovich kinetic models are shown by Equations (13)–(15).

$$q_t = q_e(1 - \exp(-k_1 t)) \quad \text{Pseudo first order} \tag{13}$$

$$q_t = \frac{k_2 q_e^2 t}{1 + k_2 q_e t} \qquad \text{Pseudo second order} \tag{14}$$

where $k_1$ and $k_2$ are the rate constants for pseudo-first-order ($h^{-1}$) and pseudo-second-order models (g $mg^{-1}$ $h^{-1}$), respectively. While $q_t$ and $q_e$ signify the adsorbed amounts of Phosphate at a given time and equilibrium ($mgg^{-1}$), respectively.

$$q_t = \frac{1}{B} \ln(AB) + \frac{1}{B} \ln(t) \quad \text{Elovich} \tag{15}$$

$q_t$ denotes sorption capacity at time t ($mgg^{-1}$), A is the initial sorption rate ($mgg^{-1}min^{-1}$), and B signifies desorption constant ($gmg^{-1}$) for a given experiment.

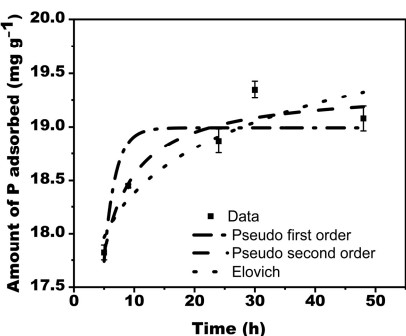

**Figure 6.** Phosphate adsorption kinetics for Douglas fir biochar treated with $MgCl_2 \cdot 6H_2O$ + KOH solutions (MKB).

Whereas the pseudo-first-order model describes reversible reactions in which equilibrium exists between the solid and liquid phases, the Elovich model defines chemical adsorption mechanism in nature, and the pseudo-second-order model often favors chemisorption processes that involve valency forces through sharing or exchange of electrons between the adsorbate and adsorbent as a covalent force.

For this study, the experimental data was best simulated by second-order kinetics ($R^2 = 0.93$), showing chemical processes-controlled adsorption. The $R^2$ corresponding to the Elovich and pseudo-first-order kinetics were 0.86 and 0.78, respectively (Table 2).

**Table 2.** MBK Phosphate adsorption Kinetic parameters.

| Model | $k_1$ ($h^{-1}$) or $k_2$ ($gmg^{-1}h^{-1}$) | $q_e$ ($mgg^{-1}$) | $R^2$ |
|---|---|---|---|
| Pseudo-first-order kinetics | 0.54 | 18.99 | 0.78 |
| Elovich | - | - | 0.86 |
| Pseudo-second-order kinetics | 0.12 | 19.36 | 0.93 |

$k_1$ ($h^{-1}$) and $k_2$ ($gmg^{-1}h^{-1}$) are the rate constants for pseudo-first order and pseudo-second order models, $q_e$ ($mgg^{-1}$) is the adsorbed amounts of phosphate at equilibrium.

The largest P adsorbed for the model and experimental data were 19.36 $mgg^{-1}$ and 19.35 $mgg^{-1}$, respectively. For proper plant growth, the sufficient concentration of available soil P required is 2 mg/g [42].

Many adsorption mechanisms responsible for phosphate sorption on metal oxide/hydroxides have been reported in the literature [43]. The probable reaction mechanisms driving phosphate sorption on MBK are ion exchange, electrostatic force of attraction, and Lewis acid–base interactions. For pH < pHpzc, phosphate sorption is primarily driven by ion exchange, in which the $H_2PO_4^-$ or $HPO_4^{2-}$ group is exchanged for the $OH^-$ group to form complexes, this occurrence is accompanied by an increase in the solution pH after adsorption due to exchange of OH- as shown by Equations (16) and (17):

$$Mg(OH)_2(s) + H_2PO_4^-(aq) \rightarrow Mg(H_2PO_4)_2(s) + 2OH^-(aq) \tag{16}$$

$$Mg(OH)_2(s) + HPO_4^{2-}(aq) \rightarrow MgHPO_4(s) + 2OH^-(aq) \tag{17}$$

In the Lewis acid–base interactions, oxygen anions of the phosphate groups are attracted to the metal active sites forming M-O coordinate bonds (Equations (18)–(20)):

$$MgO(s) + H_2O(l) \rightarrow MgO - H_2O^+(aq) \tag{18}$$

$$MgO - H_2O^+(aq) + 2H_2PO_4^-(aq) \rightarrow Mg(H_2PO_4)_2 + 2OH^-(aq) \tag{19}$$

$$MgO - H_2O^+(aq) + HPO_4^{2-}(aq) \rightarrow MgHPO_4(s) + 2OH^-(aq) \tag{20}$$

In addition to the above, the positive charge surface MBK attracts the negatively charged phosphate ion by the electrostatic force of attraction resulting in a strong bond formation. This result agrees with those reported by Takaya et al. [18].

### 3.3.3. Sorption Isotherms

Adsorption isotherms were used to quantify the phosphate sorption capacity of MBK biochar. Three models, the Langmuir, Freundlich, and Langmuir–Freundlich were employed to simulate the adsorption of phosphate on MBK (Equations (21)–(23)):

$$qe = \frac{K_L C_e Q}{1 + K_L C_e} \quad \text{Langmuir} \tag{21}$$

$$qe = K_F C_e^{\frac{1}{n}} \quad \text{Freundlich Isotherm} \tag{22}$$

$$qe = \frac{K_{L*} Q Ce^n}{1 + K_L Ce^n} \quad \text{Langmuir} - \text{Freundlich Isotherm} \tag{23}$$

where $K_L$ and $K_F$ represent the Langmuir bonding term related to interaction energies ($Lmg^{-1}$) and the Freundlich affinity coefficient ($mg^{(1-n)}$ Ln $g^{-1}$), respectively. Q denotes the Langmuir maximum adsorption capacity ($mgg^{-1}$), Ce is the equilibrium solution concentration ($mgL^{-1}$) of the sorbate, qe = the amount of solute adsorbed per gram of the adsorbent at equilibrium ($mgg^{-1}$) and, n is the Freundlich linearity constant.

The Langmuir–Freundlich adsorption isothermal equation defines the combined empirical adsorption isothermal equations of Freundlich and Langmuir. While the Freundlich model is empirical equations, the Langmuir model assumes monolayer adsorption onto a homogeneous surface with no interactions between the adsorbed molecules [44].

The models used replicated the isotherm data well (Figure 7), with $R^2$ ranging from 0.888 to 0.997 (Table 3). The experimental data fitted Langmuir–Freundlich model best ($R^2 = 0.997$), suggesting that the adsorption of phosphate on MBK is heterogeneous.

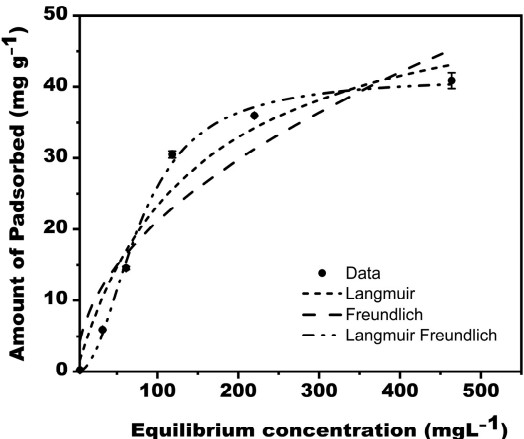

**Figure 7.** adsorption isotherm for Phosphate on MBK.

**Table 3.** MBK phosphate adsorption isotherm parameters.

| Model | $K_L$ (L/mg) | $K_F$ ($mg^{(1-n)}$ Lng$^{-1}$) | n | Q(mg/g) | $R^2$ | Adjusted $R^2$ |
|---|---|---|---|---|---|---|
| Langmuir | 0.00709 | - | - | 56.16 | 0.957 | 0.947 |
| Freundlich | - | 2.14176 | 2.01272 | - | 0.888 | 0.861 |
| Langmuir –Freundlich | $8.59 \times 10^{-5}$ | - | 2.15078 | 41.19 | 0.997 | 0.995 |

Q (mg/g) is the maximum adsorption capacity; $K_L$ (L/mg) is a Langmuir constant; $K_F$ ($mg^{1-n}$ Ln/g) is the Freundlich adsorption constant, n (dimensionless) is the adsorption affinity.

The nature of adsorption can be reflected by the value of the separation factor or equilibrium parameter $R_L$, defined by $R_L = \frac{1}{1+(1+K_L C_o)}$, where $K_L$ (L/mg) represents the Langmuir constant, and $C_o$ is the adsorbate initial concentration (mg/L). According to Foo and Hameed [44], adsorption is considered favorable if ($0 < R_L < 1$), unfavorable if ($R_L > 1$), linear for ($R_L = 1$), and irreversible.

If ($R_L = 0$) and, the lower the value $R_L$, the more favorable the adsorption is likely. From our result (Table 3), the maximum monolayer coverage capacity (Q) and the corresponding Langmuir isotherm constant $K_L$ were 56.15559 mgg$^{-1}$ and 0.00709 L/mg, respectively. The experimental separation factors ($R_L$) for different initial concentrations ranged from 0.16 to 0.49, and the $R^2$ value was 0.975, indicating that the equilibrium sorption was favorable. However, the sorption data fitted better to the Langmuir–Freundlich Isotherm model ($R^2 = 0.997$), revealing that the sorption process was heterogeneous. Furthermore, adsorption capacity can be estimated by the constant $K_F$. Whereas $\frac{1}{n}$ is a function of adsorption's strength in the adsorption process [45], it signifies heterogeneity. The larger the value of $\frac{1}{n}$, the less heterogeneity it shows, and vice versa [46]. Foo and Hameed [44] argued that chemisorption and cooperative adsorption processes are evidence if values of $\frac{1}{n}$ are less than or greater than one, respectively. For this study, the value of $\frac{1}{n}$ for Freundlich and Langmuir–Freundlich isotherms were 0.49 and 0.46, respectively (Table 3), implying that chemical processes drove the sorption of Phosphate onto MBK. The $R^2$ values corresponding to the Freundlich and Langmuir–Freundlich model in this study were 0.888 and 0.997, respectively.

### 3.4. P Retention and Availability in Soil

Batch adsorption by soil/biochar mixture was done to assess the efficiency of biochar amendments on P retention in soil. Results showed that the influence of biochar addition to soil was pH-dependent (Figure 8). At low pH (4), the sorption of P on soil biochar mixture generally improved with increasing mass of biochar added. At low pH, soil dominated by hydroxides of iron and aluminum possess a positive charge due to the association of hydrogen ions with the surface of the hydroxide groups. As a result, the hydroxyl group may be exchanged for other anions (anion exchange), for example, phosphate. A similar result was reported by Hainje [47] in which biochar addition to soil enhanced P retention in different soil types, including sandy soil, whose P retention capacity is naturally very low. This observation is also consistent with the findings of Xu et al. [48]. However, at pH (6.5), P sorption reduced as more quantity of biochar was added to the soil. Since the anion exchange is a reversible phenomenon, part of the adsorbed phosphate ion gained by soil minerals becomes readily available when soil is dominated with hydroxide ions (high pH). Therefore, as more biochar was added to the soil, the soil was more dominated by OH$^-$ due to its liming property, and consequently, more phosphate was desorbed from the soil. This finding is in accordance with those of Hovi et al. [49]. They discovered that for course texture soil (<2% clay), P adsorption diminished by increasing the mass of biochar in the soil. They ascribed the reduction in P adsorption to anionic molecules in biochar, which adsorb onto soil adsorption sites as natural organic anions resulting in competition for sorption sites, and consequently improving P availability by inhibiting PO$_4^-$ adsorption.

Retention of phosphate by soil (pH = 5.3) biochar mixture increased with increasing concentration of P added to soil biochar mixture (Figure 9). The enhanced sorption of phosphate at low soil pH can be due to the reaction of available P with hydroxides/oxides of iron and aluminum in soil [50,51], leading to the formation of less soluble compounds such as strengite (FePO$_4$ ·2H$_2$O) and variscite (AlPO$_4$ ·2H$_2$O) as shown by Equation (24). The hydroxides and oxides of metals on soil surfaces adsorbed phosphate ions because oxygen atoms of the phosphate ion donate a lone pair of electrons to fill the outer electron shell of metal atoms which are coordinated with oxygen and hydroxide ions exposed at the surfaces of soil constituents. According to Bolland et al. [51], phosphate ion replaces other anions because it is more strongly adsorbed or forms a more stable compound on soil surfaces which may later diffuse into the crystal lattice of the soil constituents. In addition,

MBK biochar had a net positive charge which attracted negatively charged phosphate ions at low pH, as indicated by pHpzc earlier in this study.

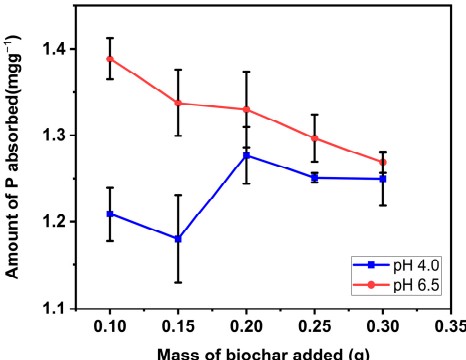

**Figure 8.** P retention and availability of soil/Douglas fir biochar treated with $MgCl_2 \cdot 6H_2O$ + KOH solutions (MKB) mixture. (Values shown are mean of n = 3, error bars shown as standard deviation).

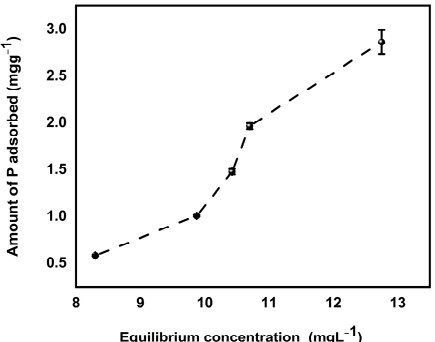

**Figure 9.** P retention of soil/Douglas fir biochar treated with $MgCl_2 \cdot 6H_2O$ + KOH solutions (MKB) mixture. (Values shown are mean of n = 3, error bars shown as standard deviation).

If soil pH is increased, however (at least 6.0), the absorbed phosphate in ($FePO_4 \cdot 2H_2O$ and $AlPO_4 \cdot 2H_2O$) can be released in the soil for plant uptake [52]. Changes in soil pH affect phosphate retention on the soil surfaces by adsorption or release of hydrogen ions. At lower pH, adsorption of hydrogen ions on the soil surface makes it more positively charged, while removing hydrogen ions from the soil surface at higher pH creates a more negative charge on the soil surface [51]. Since phosphate ions are negatively charged, they are attracted to the positively charged soil constituent at low pH and less attracted to the negative charge surface at higher pH. Also, since MBK biochar has liming property, its application in such acid soil raises soil pH, increasing the negative charge on the soil surface and making P more available for plant uptake. Similar findings were reported by Abebe [53]. However, as $H_2PO_4^-$ decreases with P uptake, strengite dissolves to maintain soluble P concentrations Equation (25).

$$
Al^{3+} + HO-\underset{\underset{OH}{|}}{\overset{\overset{O^-}{|}}{P}}=O \ + \ 2H_2O \ \rightleftharpoons \ HO-\underset{\underset{OH}{|}}{\overset{\overset{O}{\underset{|}{Al}}}{\overset{|}{P}}}=O \ + \ 2H^+ \tag{24}
$$

$$\text{Dissolved ion} \qquad\qquad\qquad\qquad \text{Complex}$$

$$FePO_4 \cdot 2H_2O + H_2O \rightleftharpoons H_2PO^{4-} + H^+ + Fe\,(OH)^3 \tag{25}$$

## 4. Conclusions

The current study evaluated the P availability of p-enriched modified Douglas fir biochar for probable practicality as an alternative slow-release P fertilizer. Before sorption, the point of zero charges of Douglas fir modified biochar was determined. Results showed that phosphate sorption was favored at a pH less than 10.3, implying that for soil with low pH, P availability would be reduced.

In each case, three models were used to simulate reaction kinetics and isotherm. Results indicated that the sorption of P on MBK was heterogeneous and driven by chemical processes. The maximum P adsorbed corresponding to the best model was 41.19 mg g$^{-1}$. The smallest and sufficient available soil P concentration required for plant growth has been estimated at 0.0002 mg g$^{-1}$ [54] and 2 mg g$^{-1}$ [42], respectively.

P sorption on biochar soil mixture was pH dependent. At lower pH = 4, P sorption improved with increasing mass of biochar added. However, at pH = 6.5, less P was adsorbed as more biochar was added to the soil. At low pH, less P would be available for plants as sorption of P on biochar and soil/biochar would be favored. In contrast, at pH close to neutral, less P is adsorbed and can be available for plants. Therefore, for very low pH, the use of P-enriched Douglas fir biochar should be accompanied by liming to increase the pH and hence P availability. More studies should be done to validate the practicability of P-enriched Douglas fir biochar as soil amendments.

**Author Contributions:** Conceptualization, B.A. and T.M.; methodology, B.A. and C.N.; software, C.N. and B.A.; validation, C.N., B.A. and N.K.D.; formal analysis, B.A., C.N., N.K.D. and A.H.; investigation, B.A.; resources, T.M.; data curation, B.A. and C.N.; writing—original draft preparation, B.A.; writing—review and editing, B.A. and C.N.; visualization, B.A.; supervision, T.M.; project administration, T.M.; funding acquisition, T.M. All authors have read and agreed to the published version of the manuscript.

**Funding:** This research received no external funding.

**Data Availability Statement:** The data that support the findings of this study are available from the corresponding author upon request.

**Acknowledgments:** Department of Chemistry Mississippi State University, Fulbright organization, and US Department of State for the scholarship awarded to B.A.

**Conflicts of Interest:** The authors declare no conflict of interest.

## Abbreviations

| | |
|---|---|
| ATR-FTIR | Attenuated total reflectance Fourier Transformed Infrared Spectroscopy |
| BET | Brunner–Emmet–Teller |
| DFB | Douglas fir biochar |
| EDX | Energy dispersive spectroscopy |
| MB | Magnesium chloride modified Douglas fir biochar |
| MBK | Magnesium chloride plus potassium hydroxide modified Douglas fir biochar |
| PEM | P-enriched biochar (post sorption MBK) |
| pHpzc) | pH point of zero charge |
| SEM | Scanning electron microscopy |
| US EPA | United States Environmental Protection Agency |
| XRD | X-ray diffraction |

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
