# Peer review of "Sorption of Phosphate on Douglas Fir Biochar Treated with Magnesium Chloride and Potassium Hydroxide for Soil Amendments"

_processes, doi:10.3390/pr11020331_

Round 1
Reviewer 1 Report
It is a straightforward work and contribution. I have no comments.
Author Response
Thank you for reviewing the manuscript and for the positive comments
Reviewer 2 Report
Remarks:
Despite an interesting issue, the article was written in a not very professional and not very careful manner:
A very modest introduction should be significantly developed.
There is no information in the methodology, based on how many samples the tests were performed, no information on what type of SEM microscope was used, what the samples were sprayed on ...
The entire article does not meet the formal requirements, the formatting does not comply with the requirements in mdpi journals.
Wrong way of citation.
Figures - hardly legible, it is not known what scale is in the photo SEM. In addition, there is no information on what is what, e.g. figure 2 is unacceptable, no vertical scale in the spectra, no information whether it is a point analysis, if so why is it not marked in the photo?
Format, not acceptable; no division into subsections, sometimes the authors write a figure in the text, sometimes a figure, no explanation of the abbreviations used, no citation in the text of the reactions listed.
Conclusions or summary discussing the effect of pH on sorption capacity are nothing new, we have to agree with the authors that more research should be done.
Literature should include more recent items and its method of citation in the text is also incorrect.
The article may have scientific value in the future, but it is unacceptable in its current form.
Author Response
Thank you for your valuable comments. The manuscript format has been revised to suit the requirements for mdpi journals. Figure 2 has been replaced with a clearer one as suggested by the reviewer, specifications for SEM instrument are indicated in document (see line 125 to 128). Further, we agree with the reviewer that many similar studies have been reported on pH effects on sorption capacity. However, P retention on biochar is affected by many factors including the nature of the feedstock. For this study, Douglas fir biochar was used because of its low-cost and other environmental benefits. Most studies done concentrated on the use of agricultural residues such as rice hulls and chicken manure. Commercial biochar such as Douglas fir biochar have not been widely studied for soil amendments despite its availability (waste). The use of Douglas fir biochar can be a useful strategy for waste management, carbon sequestration, reducing risk for water contamination as well as the need for application of commercial phosphate fertilizer and lime. More broadly, the United Nations Intergovernmental Panel on Climate Change (IPCC) report of August 2021 forecast that biochar should be sequester in huge quantity in soil (109 tons/yr) to combat the growth of atmospheric CO2 while improving the soil for production. Therefore, for improved P retention, Douglas fir biochar is an important material to study
Reviewer 3 Report
The manuscript presents results of a study aimed at evaluating the retention and availability of P by modified Douglas fir biochar and assessing its potential use as a soil amendment. The study has been well designed and carried out. Results were correctly described and interpreted. Conclusions are sound and supported by presented data. The text is well structured and easy to follow. I do not have any important comments to the text. There are some editorial mistakes (eg. lines 48, 188 “tal” instead of “al”; line 184 3 should be in upper index) and the first sentence of abstract is so general that it could be added to nearly every scientific paper dealing with arable soils but they are of minor importance.
Author Response
Reply to reviewer 3’s general comment
We appreciate the positive feedback from the reviewer. Thank you for the careful and helpful suggestions.
Reviewer 3’s comment 1
There are some editorial mistakes (eg. lines 48, 188 “tal” instead of “al”; line 184 3 should be in upper index)
Reply to reviewer 3’s to comment 1:
Thank for the suggestions. Corrections has been made in lines 48, 188 and 184 as suggested by the reviewer.
Reviewer 3’s comment 2
The first sentence of abstract is so general that it could be added to nearly every scientific paper dealing with arable soils, but they are of minor importance
Reply to reviewer 3’s comment 2:
Thank for your comment. We have improved the first sentence of the abstract. The new sentence now reads “With increasing climate variability, sustainable crop production approach remains an indispensable concern across the globe’’ instead of “The need for environmentally sustainable crop production approach remains an indispensable concern across the globe’’
Reviewer 4 Report
The paper presents a study on the enrichment of Douglas fir biochar for use as an additive supplier of phosphorus to farm soils, in a perspective to decrease the consumption of fertilizers.
The experimental characterization and sorption studies are well-structured and correctly planned. However, given the heterogeneities of the different types of soils, a characterization of the soil used in the sorption tests with and without biochar blend was desirable.
The manuscript has some typos that can easily be corrected:
1 - The English writing deserves a careful revision, as it has some grammatical (singular/plural) agreement errors, for example in line 268 "Three models was used".
2 - The chemical symbol for phosphorus sometimes appears in lowercase (p instead of P, eg. in lines 167 and 349).
3 - The size and font in chemical equations are very different from normal text.
4 – Line 106 - has an extra period.
5 – Lines 164 and 174 mention that it was previously described in 2.4.2, but the text has no such numbering.
6 – Line 225 - typed quart instead of quartz.
Author Response
Reply reviewer 4’s general comment
Thank you for your positive comments and suggestion to improve the manuscript. We agree with the reviewer that soil is heterogeneous, and characterization of the soil used in the sorption test would be vital. However, given the limitation that Beatrice Arwenyo the first author has left United State for her Country, this test may not be possible for this study. We believe that the result obtained in this study has sufficiently demonstrated that biochar can improve P retention in sandy soil. The effects of various conditions could feasibly be included in the study of different soil type later.
Reviewer 4’s comment 1
The English writing deserves a careful revision, as it has some grammatical (singular/plural) agreement errors, for example in line 268 "Three models was used".
Reply to reviewer 4’s comment 1
Thank for your comment. Grammatical errors were corrected as suggested
Reviewer 4’s comment 2
The chemical symbol for phosphorus sometimes appears in lowercase (p instead of P, eg. in lines 167 and 349).
Reply to reviewer 4’s comment 2
Thank you for the useful comment to improve the manuscription. The symbol for phosphorus has been changed from ‘‘p’’ to ‘’P’’ in lines 167 and 349.
Reviewer 4’s comment 3
The size and font in chemical equations are very different from normal text.
Reply to reviewer 4’s comment 3
Thank you for the valuable comment. The font size in chemicals equations and normal text has been corrected.
Reviewer 4’s comment 4
Line 106 - has an extra period
Reply to reviewer 4’s comment 4
Thank for your comment. The extra period in line 106 has been deleted
Reviewer 4’s comment 5
– Lines 164 and 174 mention that it was previously described in 2.4.2, but the text has no such numbering.
Reply to reviewer 4’s comment 5
Thank you for your useful comments. Section numbers for the entire manuscript have been included.
Reviewer 4’s comment 6
Line 225 - typed quart instead of quartz.
Reply to reviewer 4’s comment 6
Thank for your comment. The typing error in line 225 has been corrected. The word ‘quart’ was replaced with “quartz “
Round 2
Reviewer 2 Report
The authors referred to earlier comments.One has to agree with the authors that more analyzes
are needed to confirm the practicality of the conducted research.
At the end of the article, it was advisable to make a table explaining
the abbreviations used in the work.
Author Response
Response is attached
